# Immunohistochemical Analysis of DNA Repair- and Drug-Efflux-Associated Molecules in Tumor and Peritumor Areas of Glioblastoma

**DOI:** 10.3390/ijms22041620

**Published:** 2021-02-05

**Authors:** Cristiana Angelucci, Alessio D’Alessio, Silvia Sorrentino, Filippo Biamonte, Umberto Moscato, Annunziato Mangiola, Gigliola Sica, Fortunata Iacopino

**Affiliations:** 1Dipartimento di Scienze della Vita e Sanità Pubblica, Sezione di Istologia ed Embriologia, Università Cattolica del Sacro Cuore-Fondazione Policlinico Universitario “Agostino Gemelli”, IRCCS, 00168 Rome, Italy; cristiana.angelucci@unicatt.it (C.A.); silvia.sorrentino@unicatt.it (S.S.); gigliola.sica@unicatt.it (G.S.); fortunata.iacopino@unicatt.it (F.I.); 2Dipartimento di Scienze Biotecnologiche di Base, Cliniche Intensivologiche e Perioperatorie, Istituto di Biochimica e Biochimica Clinica, Università Cattolica del Sacro Cuore, 00168 Rome, Italy; filippo.biamonte@unicatt.it; 3Dipartimento di Scienze di Laboratorio e Infettivologiche, Unità Operativa Complessa di Chimica, Biochimica e Biologia Molecolare, Università Cattolica del Sacro Cuore-Fondazione Policlinico Universitario “Agostino Gemelli”, IRCCS, 00168 Rome, Italy; 4Dipartimento di Scienze della Vita e Sanità Pubblica, Sezione di Medicina del Lavoro e Igiene di Sanità Pubblica, Università Cattolica del Sacro Cuore-Fondazione Policlinico Universitario “Agostino Gemelli”, IRCCS, 00168 Rome, Italy; umberto.moscato@unicatt.it; 5Dipartimento delle Scienze della Salute della Donna, del Bambino e di Sanità Pubblica, Fondazione Policlinico Universitario “A. Gemelli”, IRCCS, 00168 Rome, Italy; 6Unità Operativa Complessa di Neurochirurgia, Ospedale Santo Spirito, 65124 Pescara, Italy; annunziato.mangiola@unich.it; 7Dipartimento di Neuroscienze, Imaging e Scienze Cliniche, Università “G. D’Annunzio”, 66013 Chieti, Italy

**Keywords:** glioblastoma, peritumoral area, MGMT, BCRP1, A2B5, chemotherapy, cancer stem cells

## Abstract

Glioblastoma (GBM), the most commonly occurring primary tumor arising within the central nervous system, is characterized by high invasiveness and poor prognosis. In spite of the improvement in surgical techniques, along with the administration of chemo- and radiation therapy and the incessant investigation in search of prospective therapeutic targets, the local recurrence that frequently occurs within the peritumoral brain tissue makes GBM the most malignant and terminal type of astrocytoma. In the current study, we investigated both GBM and peritumoral tissues obtained from 55 hospitalized patients and the expression of three molecules involved in the onset of resistance/unresponsiveness to chemotherapy: O6-methylguanine methyltransferase (MGMT), breast cancer resistance protein (BCRP1), and A2B5. We propose that the expression of these molecules in the peritumoral tissue might be crucial to promoting the development of early tumorigenic events in the tissue surrounding GBM as well as responsible for the recurrence originating in this apparently normal area and, accordingly, for the resistance to treatment with the standard chemotherapeutic regimen. Notably, the inverse correlation found between MGMT expression in peritumoral tissue and patients’ survival suggests a prognostic role for this protein.

## 1. Introduction

Glioblastoma (GBM) is a grade IV tumor of the central nervous system (frequently referred to as a grade IV astrocytoma). GBM is the most common and malignant intracranial tumor in adults [1]. Current treatment choices for GBM at diagnosis are limited, and standard treatment consists of maximal surgical resection, followed by concurrent radiation therapy and chemotherapy [2]. Despite the massive surgical resection, GBM preserves high invasiveness with frequent recurrence, allegedly due to infiltrating tumor cells remaining in the adjacent brain tissues, leading to poor prognosis characterized by a median overall survival ranging from 12 to 15 months [3]. Progress has been made in the development of prognostic tools, identifying molecular markers that help predict survival and conceive individualized therapy regimens, improving the response rate [4]. However, promising approaches aimed at targeting primary and recurrent GBM, such as immunotherapies, receptor tyrosine kinase inhibitors, targeted toxins, and anti-angiogenic drugs, resulted in unsatisfactory outcomes [5]. Notably, due to its involvement in the progression and recurrence of GBM, the peritumoral (PT) tissue is emerging as a remarkable field of investigation in this area. Our previous studies have focused on the potential role of the GBM PT tissue in recurrence, taking into account the tissue surrounding the enhanced lesion up to 3.5 cm from the tumor margin [6,7,8,9,10,11]. Within the GBM neighboring tissue, the presence of reactive astrocytes and microglia [8,9,10,11,12], abnormal gene expression [10,11,12,13], and altered tumor-like neovascularization were assessed [9]. Evidence has suggested the presence of a subpopulation of cells endowed with a cancer-stem-like cell (CSC) phenotype [7,8,9,11,14] residing in the PT tissue that may be responsible for the occurrence of pre-/pro-tumorigenic events and for the resistance to chemo- and radiotherapy [15,16]. GBM recurrence in the PT area has been observed approximately in 90% of patients and appears to be tightly correlated with the resistance to the standard temozolomide (TMZ) chemotherapy [17]. TMZ is an alkylating agent that modifies DNA by introducing methyl groups at the N7 position of guanine, the O3 position of adenosine, and the O6 position of guanosine, leading to DNA breaks and subsequent apoptosis of tumor cells [18]. Overexpression of the dealkylating enzyme O6-methylguanine methyltransferase (MGMT), which removes the cytotoxic methyl groups from the O6 position, has been suggested to play a crucial role in the development of TMZ resistance as well as a major cause of standard therapy failure [19,20]. Methylation of specific promoter sites in the MGMT promoter decreases protein expression, making promoter methylation status a predictive biomarker in response to TMZ [20]. Failure of both conventional and targeted therapy can also occur through an active efflux of drugs, leading to their decreased intracellular levels. Drug insensitivity consequently develops, usually to various agents. One of the well-defined sources of this cancer cell multi-drug resistance (MDR) is the increased expression of ATP binding cassette (ABC) transporters, which extrude chemically unrelated drugs out of the cell [21]. Among them, breast cancer resistance protein (BCRP1), also known as ABCG2, one of 48 human transporters of the ABC family, is found in normal brain tissue at the blood–brain barrier, where it likewise plays a protective function against potentially damaging compounds [22]. It has been demonstrated that efflux transport in the blood–brain barrier is involved in limiting the brain distribution of palbociclib, with critical implications for determining effective dosing regimens of palbociclib therapy in the treatment of brain tumors [23]. Moreover, it has been demonstrated that BCRP1 function not only correlates with high stem cell (SC) marker expression and self-renewal capacity but can also actively drive these SC features in some GBM cultures without regulating glioma cell tumorigenicity or radiation resistance [24]. The strong TMZ resistance of CD133+ CSCs obtained from GBM patients is probably contributed by the CD133+ cells displaying high BCRP1 and MGMT levels [25]. Several gangliosides, including GD2 and GD3, have been identified as GBM CSC markers playing a key role in GBM tumorigenesis and are now being considered attractive therapeutic targets [26,27]. Notably, it was recently demonstrated that anti-GD2 monoclonal antibody immunotherapy significantly increased TMZ toxicity in cultured GBM cells by enhancing TMZ uptake and prevented the extension of the TMZ-resistant CSC pool within the tumor bulk in vivo [28]. The A2B5 epitope belongs to the sialoganglioside family and is expressed by GBM CSCs displaying a high proliferation rate and strong migratory/invasive capacity and able to originate tumors in nude mice independently of CD133 expression [29]. A2B5 is thought to be a marker of poor prognosis, and low-grade A2B5+ astrocytomas may have a higher rate of recurrence than the tumors of A2B5- lineage [30]. In this study, a comparative immunohistochemical analysis of the expression of MGMT, BCRP1, and A2B5 was performed on paraffin-embedded tissue sections obtained from GBM and PT parenchyma of 55 patients diagnosed with primary supratentorial GBM who underwent en bloc surgery. The resulting PT profile was compared with that of the GBM zone to identify characteristics specific to the PT zone and its correlation with clinical parameters.

## 2. Results

### 2.1. Immunohistochemical Analysis of MGMT, BCRP1, and A2B5 in GBM and PT Tissue

The expression of MGMT, BCRP1, and A2B5 was analyzed by immunohistochemistry, both in GBM and in PT tissue obtained by surgery. Immunoreactivity of the three molecules was found in both GBM- and PT-tissue-derived sections (Figure 1). Due to the great histological complexity of the tumor and the PT tissue, we performed an unbiased stereometric analysis of histological samples by evaluating the cell density of both positive and negative cells for MGMT, BCRP1, and A2B5, as described in the Materials and Methods section. This analysis clearly showed a higher cell density in GBM than in PT tissue (Figure 1). As regards MGMT expression in GBM samples, it turned out to be mainly localized in the nuclei of endothelial cells (ECs) of the neo-vasculature (Figure 1A). Yet, we did not detect MGMT expression in malignant cells (i.e., cells with atypical nuclei). In PT tissue, MGMT was highly expressed both in ECs and all around the vessel wall (Figure 1B). The results of the stereometric analysis clearly showed that the cell density of MGMT-negative cells was higher in tumor samples compared to the PT area (*p* < 0.0001, two-sample Wilcoxon rank-sum, Mann–Whitney test; PT: median, 52.1; range, 0.54–92.4; GBM: median, 10.2; range, 0.0–76.5) (Figure 2A). Immunohistochemical analysis of BCRP1 indicated that the molecule was expressed in the plasma membrane and within the cytoplasm both in GBM and in PT tissue (Figure 1C,D). In addition, stereometric analysis performed on histological samples indicated a significantly higher BCRP1-positive cell density in PT tissue than in GBM (*p* < 0.02; PT: median, 18.1; range, 0.4–95.6; GBM: median, 6.9; range, 0.0–72.1) (Figure 3A,B). As it concerns the expression of A2B5, immunohistological analysis revealed that both in GBM and in PT tissue, the ganglioside was located on the cell surface and within the cytoplasm (Figure 1E,F). In addition, A2B5 immunoreactivity was found in ECs and in other cell types in the surrounding tissue. However, there was no statistically significant difference in A2B5 expression between PT tissue and GBM (*p* = 0.69; PT: median, 28.4; range, 0.0–98.4; GBM: median, 17.3; range, 0.0–96.8) (Figure 4A,B).

### 2.2. Survival Analysis of GBM Patients

The correlation of MGMT, BCRP1, and A2B5 expression with prognosis of glioma patients was investigated through Kaplan–Meier survival curve analysis with a log-rank test of 50 GBM patients (Table 1, Figure 5). To determine the prognostic value of MGMT, BCRP1, and A2B5 in GBM patients, univariate Cox regression analysis was employed. Remarkably, only high MGMT expression in the PT area correlated with a high-risk factor for GBM patients (*p* < 0.05; *z* = −2.12; 95% confidence interval 0.026–0.00095). According to the Cox model (proportional hazard, Cox regression: Breslow method for ties), the variable significantly correlated with survival was the PT histochemical group, which has a negative *z* value to indicate that its expression is inversely correlated with survival (Figure 5). The ROC curve did not identify any cutoff that can be used to define positive or negative samples. In addition, based on our statistical analysis, it is conceivable that the decrease in the average age of the patients (<59 years) as well as in MGMT expression (<50%) would increase the survival time of some of the subjects analyzed without any significant gender difference, although over 60% of them would have died within 20 months from histochemical diagnosis. Moreover, with an increase >50% of MGMT expression and an average age < 59 years, 50% of the patients survived after 30 months from diagnosis. Therefore, the only parameter that significantly correlated with survival was the age. Patients ≤ 64 years old at diagnosis had a better median survival time compared with patients diagnosed at ≥65 years of age (14 and 12 months, respectively; *p* = 0.028 log-rank test).

## 3. Discussion

GBM is among the deadliest of gliomas, characterized by high heterogeneity and elevated resistance to treatment with TMZ, a cytotoxic drug employed to block proliferation of tumor cells. A number of studies have suggested that both intrinsic and acquired TMZ resistance are related to the expression level of DNA alkylating proteins and DNA repair enzymes [19,20]. Molecular subtyping of GBM tissue has shown promise in recognizing subsets of patients who may be uniquely responsive to specific adjuvant therapeutic approaches [31]. Indeed, aggressive surgery and standard multimodal adjuvant treatments fail to prevent local recurrence that invariably appears within the surrounding brain tissue [32]. Therefore, there is an urgent need to deepen our understanding of the histological and molecular features of the seemingly normal PT tissue to investigate the mechanisms responsible for the onset of relapses, tumor progression, and treatment failure in GBM patients. In GBM, promoter methylation of the gene encoding MGMT is the molecular fingerprint with the highest clinical impact. The MGMT gene codes for a ubiquitously expressed nuclear DNA repair protein participating in cellular defense against mutagenesis and toxicity due to alkylating agents such as TMZ [33]. The intracellular level of MGMT has been predicted to correlate with chemoresistance. The methylation status of the MGMT promoter is, to date, considered of high clinical interest and represents a master mechanism accountable for chemoresistance, although the occurrence of an abnormal DNA repair system and multi-drug resistance contribute to this scenario [34]. Randomized trials of newly diagnosed GBM patients treated with TMZ have revealed that MGMT gene silencing by promoter hypermethylation correlates with improved survival. On the other hand, GBM patients with an unmethylated MGMT promoter, leading to high MGMT protein expression, show poorer outcomes and a negligible response to TMZ [3,7]. Notably, in these latter GBM patients, various microRNAs (miR-142-3p, miR-181d, miR-221/222, miR-370-3p, and miR-603) acting as epigenetic negative regulators of MGMT expression and/or functionality are able to increase sensitivity to TMZ in vitro as well as in vivo [35]. In this study, we found a significantly higher percentage of cells showing positive staining with the antibody to MGMT in PT tissue than in GBM, independently from the presence of tumor cells in the former area. In addition, in both GBM and PT, MGMT expression was not detected in malignant cells, and in PT tissue the enzyme was clearly expressed in ECs of the neo-vasculature and around the vessel wall. These findings are in line with immunohistochemical data obtained from a medulloblastoma study where adjacent normal PT cerebellar tissue did not show immunoreactivity for MGMT, while ECs of this area were decorated by the antibody [36]. The role of MGMT in modulating GBM-induced angiogenesis and response to anti-angiogenic drugs has been also established [37]. Indeed, it has been demonstrated that GBM cells stably transfected with MGMT have a decreased capability to induce tube formation in vitro and tumorigenicity in vivo compared to mock transfected cells. The possible confirmation of these results in vivo could involve the evaluation of the expression levels of MGMT as a discriminant for selecting patients who may respond to anti-angiogenic therapy. In our study, unlike ECs, no tumor cells in GBM or PT tissue were stained for MGMT. In this setting, the shift toward the described anti-angiogenic profile would not occur. Indeed, we found that a higher percentage of MGMT-positive cells in the PT area correlates with poorer patient survival. Our findings suggest that an MGMT-positive endothelium and the perivascular niche of the PT vessels may provide a protective environment against TMZ. This seems to be already established in GBM tissue. Evidence has actually suggested that GBM-associated ECs and the perivascular niche make a significant contribution to TMZ resistance [38,39]. In vitro studies have demonstrated that although TMZ is highly cytotoxic to the MGMT-negative GBM cell line U87MG, it does not affect the cell viability, proliferation rate, or migration of GBM-associated ECs [40]. Likewise, data obtained from an in vivo mouse intracranial GBM model indicated no significant effects induced by TMZ on microvessel density. Overexpression of the CSC-associated functional markers of the superfamily of ABC transporters represents one of the main mechanisms of drug resistance in the brain, leading to poor drug delivery to tumor cells [21]. Just as for MGMT expression, we found a higher percentage of cells positively stained for the ABC transporter BCRP1 in PT tissue with respect to GBM samples. This finding is in line with recently reported evidence concerning the expression of another member of the superfamily of the ABC transporters, ABCC3, in human glioma samples. The expression level of ABCC3 in glioma tissues was significantly lower than that in normal brain tissues [29]. Although ABCC3 function has not yet been elucidated, its downregulation has been found to be linked to patients’ overall survival and disease-free survival rates [29]. This makes it conceivable for the ABCC3 gene to play a prognostic role. As for A2B5, it has been found that the expression level of this ganglioside in different glioma cell lines positively correlates with cell proliferation, migration, clonogenicity, and tumorigenicity [41]. The A2B5-positive cell fraction isolated from GBM was reported to display morphological and functional features typical of brain-tumor-initiating cells in terms of the ability to migrate, proliferate, and differentiate into oligodendroglial and type-1/type-2 astroglial cells [42], while A2B5-negative cell fraction was able to form tumors when injected into nude mice [43]. Just as for MGMT and BCRP1 expression, we found a higher density of cells showing immunopositive staining for A2B5 in the tissue adjacent to GBM with respect to the malignant lesion. The presence of cells with stem-like cell features in the tissue surrounding GBM has already been established. We previously reported the expression of the progenitor/stem cell marker GD3 ganglioside in GBM, PT tissue, and CSCs, isolated from the tumor and PT areas [11]. We found GD3 expression in PT tissue up to a remarkable distance of 3.5 cm from the tumor margin, despite the absence of cells with morphological features of malignant transformation. Moreover, we have shown GD3 expression in ECs of both tumor and PT tissue, suggesting that this ganglioside is also involved in neoangiogenesis. Indeed, gangliosides have been found to induce both VEGF secretion by GBM cells [44] and EC migration [45]. In our previous studies, the expression of phosphorylated extracellular signal-regulated kinases 1/2 (pERK1/2), phosphorylated c-Jun NH2-terminal kinases (pJNK), other stem cell markers (such as Nestin, CD133, and NG2 proteoglycan), as well as angiogenesis-related factors (CD105, VEGF, VEGFR1/2, HIF-1α, and HIF-2α), has been demonstrated in GBM and PT tissue, even in the absence of frankly neoplastic cells [7,8,9,11,46,47]. Moreover, genome-wide expression profiles of samples isolated from the GBM tumor mass and apparently healthy white matter adjacent to the tumor revealed various dysregulated genes in this latter area [10]. Thus, the results of the current study enrich a growing body of knowledge testifying to the occurrence of early tumorigenic events in the tissue surrounding GBM. Also, we cannot exclude that the existence of CSCs in such an altered microenvironment might be responsible for tumor recurrence as well as for resistance to standard therapy. In conclusion, we believe that the detection of high levels of the three above-mentioned molecules, engaged in DNA repair and drug efflux in the apparently healthy GBM neighboring tissue, may represent a first line of resistance to treatment with the standard chemotherapeutic regimen. Moreover, the inverse correlation found between MGMT expression in PT tissue and patients’ survival suggests a prognostic role for the immunodetection of this protein in the GBM neighboring tissue.

To better elucidate the significance of these data, further coexpression analysis of markers of stemness and differentiation would be important to define the single-cell-type relationship with tumor progression and response to therapy.

## 4. Materials and Methods

### 4.1. Patients and Tissue Samples

Primary tumor specimens and adjacent brain tissue were obtained from 55 patients diagnosed with primary supratentorial GBM who underwent en bloc surgery at the Institute of Neurosurgery of the Catholic University of the Sacred Heart, Rome, Italy. Neuronavigation and intraoperative ultrasound were used to define and maximize the extent of intracranial tumor resection. The tumor and neighboring apparently normal tissue were removed en bloc. The surgical specimens were cut and opened in a bookwise fashion. Using this technique, the difference between the tumor border and its surrounding apparently normal white matter was evident and the distance from the white matter adjacent to the tumor edge and this latter was well defined and measured [48]. In each patient, complete removal of the enhanced lesion (T1-weighted zone) and extension of the resection till the T2-weighted magnetic resonance imaging (MRI) zone were confirmed by early contrast MRI (within 24/48 h after surgery), comparing pre- and post-operative contrast-enhanced images. Paired GBM and PT tissue samples were obtained from the enhanced lesion without areas of necrosis (GBM) and from the white matter at a distance of <1 cm from the macroscopic tumor border (PT tissue). None of the patients had received radiotherapy or chemotherapy before surgical resection. Thirty-five to forty days after surgery, all patients underwent irradiation and chemotherapy was performed according to the literature [2]. The histological diagnosis was evaluated for sections stained with hematoxylin-eosin according to the 2007 World Health Organization (WHO) classification guidelines [1]. Neoplastic cells were identified by large hyperchromatic and irregular nuclei, while reactive astrocytes were recognized by the dendritic morphology of their abundant eosinophilic cytoplasm and large, eccentric nuclei, according to Hoelzinger et al. [49]. The survival period was defined as the time from the date of surgery to the date of death.

### 4.2. Immunohistochemistry and Stereological Analysis

Five-micrometer-thick sections of 10% formalin-fixed, paraffin-embedded tissues from tumor and PT samples were used for immunohistochemistry analysis. Before proceeding with immunolabeling, sections were deparaffinized in xylene and treated with 0.3% hydrogen peroxide in methanol for 20 min to block endogenous peroxidase activity. The sections were washed in phosphate-buffered saline and then incubated with primary antibodies (dilution 1:50) overnight, followed by incubation with the appropriate secondary antibodies. The substrate chromogen, 3.30-diaminobenzidine (DAB), enabled visualization of the nuclear expression of MGMT via a brown precipitate. Hematoxylin (blue) counterstaining enabled the visualization of the cell nuclei. Omission of the primary antibody served as a negative control. Immunostaining for MGMT, BCRP1, and A2B5 was performed using the standard protocol on automated staining Dako Autostainer PlusLink (Dako, Glostrup, Denmark) until antigen unmasking. Target retrieval solution (pH 9) was used for antigen unmasking on the Dako PT Link Autostainer. Then a manual staining method was used to reveal protein expression. Hydrogen peroxide (0.3%) was applied to block endogenous peroxide activity. A block of non-specific staining (Super Block, UCS Diagnostic S.r.l., Morlupo, Italy) was followed by incubation with primary antibodies, anti-MGMT (1:200), anti-BCRP1 (1:100), and anti-A2B5 (1:100), overnight at 4 °C. Subsequently, the sections were incubated with an HRP/Fab polymer conjugate (SuperPicTure Polymer DetectionKit, Invitrogen, Camarillo, CA, USA). The reaction was visualized with 3,3′-diaminobenzidine (Peroxidase DAB substrate kit, Vector Laboratories Inc., Burlingame, CA, USA), resulting in the expected brown-colored signal. Finally, all the sections were counterstained with Mayer’s hematoxylin and dehydrated in ethanol before mounting. No staining was detected in negative control sections in which the primary antibody had been omitted. For accurate quantification of immunopositive cells, an unbiased stereological technique by means of the Stereo Investigator system was employed (Stereo Investigator software, v 9.14 2010, MicroBrightField Europe, Magdeburg, Germany), as previously reported [47,50,51,52]. A stack of MAC 6000 controller modules (Ludl Electronic Products, Ltd., Hawthorne, NY, USA) was configured to interface a light microscope (Nikon Eclipse 80i, Nikon Corporation, Tokyo, Japan) with a motorized stage and a color digital camera (MicroBrightField) with a PC workstation. First, each region of interest (ROI) was outlined at low magnification (×100) and then scanned using the Meander scan function. All immunopositive and immunonegative cells within the ROI were counted at x400 magnification. The cell density was determined by dividing the number of immunopositive or immunonegative cells by the area of the ROI (cells/mm^2^). Data expressed as a percentage of positive cells were used for statistical analysis. It is worth mentioning that due to the limited amount of peritumoral tissue available in some cases, only 50 of the 55 patients were analyzed for MGMT, BCRP1, and A2B5.

### 4.3. Statistical Analysis

Patient variables were collected in Excel sheet 2011 for Mac Rev. 14.7.7 and then analyzed with Stata/IC 14.2 for Mac software Rev. 29 January 2018 (StataCorp Lakeway, College Station, TX, USA). Kaplan–Meier life-table analysis was performed for actuarial survival analysis. Continuous variables were reported as the mean and standard deviation or median (range), and categoric variables were reported as frequencies (%). All variables, including demographic ones, were normally distributed and statistically significant with Shapiro–Wilk and Shapiro–Francia tests for normality, except for the KPS variable. The survival time was considered the end point, the nonparametric variables were compared with the Pearson *χ*^2^ test, and all the remaining variables were compared by parametric tests such as analysis of variance and the *τ*-test. Univariate analyses (nonparametric log-rank test of equality across strata and Cox proportional hazard regression as a semiparametric model) were performed to identify proportionality between groups and to test associations between survival and the variables of interest, which were useful in designing the model of the survival curve. Variables with univariate *p*-values of 0.1 were considered in multivariable models, which were also adjusted for age and gender. To facilitate the interpretation of hazard and odds ratios, continuous variables were dichotomized in the final models, based on the cutoff point as determined by receiver operating characteristic (ROC) analysis. Stepwise backward elimination (SBE) is a procedure that helps select important factors and eliminate weak factors; therefore, multivariate logistic regression (MVREG) and SBE procedures were performed to develop the final models. A *p*-value of 0.05 was considered statistically significant for all analyses. The long-term survival analyses were calculated from the date of the histochemical diagnosis.

## Figures and Tables

**Figure 1 ijms-22-01620-f001:**
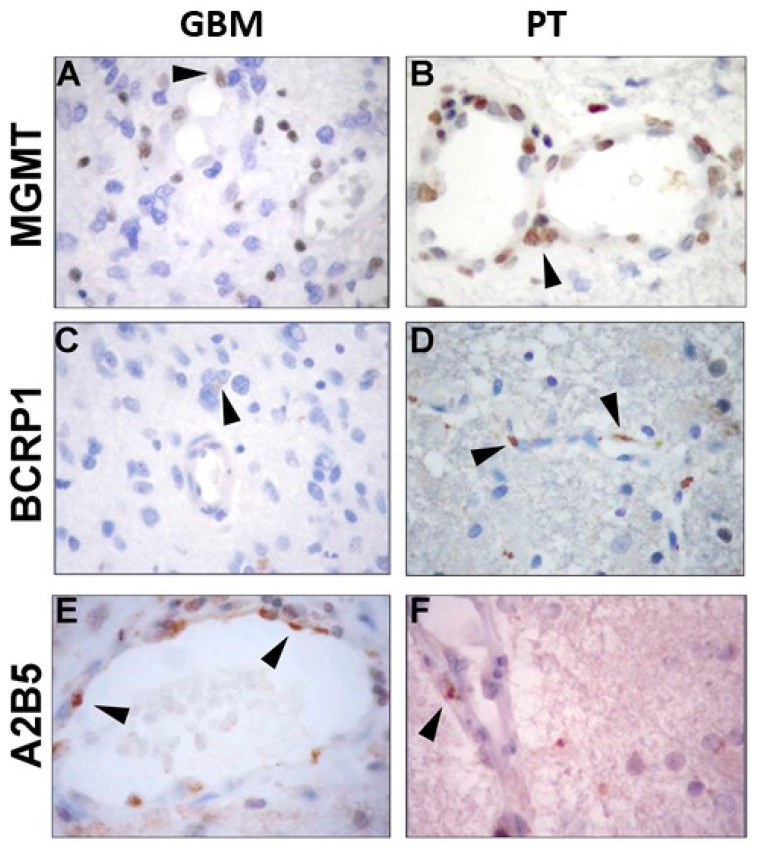
Immunohistochemistry analysis of glioblastoma (GBM) and peritumoral (PT) areas of surgical samples. Primary tumor specimens and adjacent brain tissue were collected, fixed in 10% formalin, and analyzed by means of immunohistochemistry. Arrowheads indicate the immunoreactivity of O6-methylguanine methyltransferase (MGMT), breast cancer resistance protein (BCRP1), and A2B5 in both areas. (**A**,**C**,**E**) MGMT, BCRP1, and A2B5 expression in GBM samples; (**B**,**D**,**F**) MGMT, BCRP1, and A2B5 expression in PT tissue.

**Figure 2 ijms-22-01620-f002:**
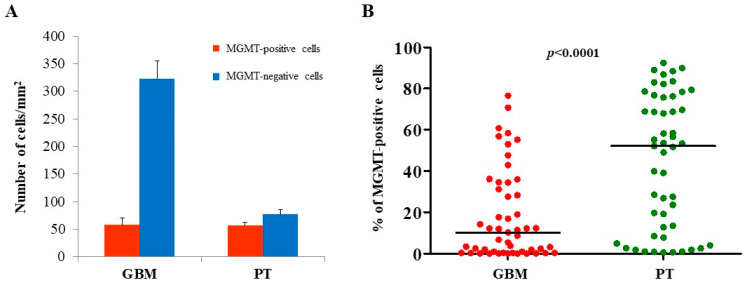
Stereological analysis of MGMT expression in both GBM and PT areas. Histological samples subjected to immunohistochemistry analysis were further evaluated by stereological analysis. The number of both labeled and unlabeled cells was employed to quantify the expression of MGMT in both GBM and PT tissue, as specified in the Materials and Methods section. (**A**) Red and blue bars show the number of MGMT-positive and MGMT-negative cells/mm^2^, respectively. The data are presented as the mean ± standard error (SE). (**B**) Plots show the percentage of MGMT-positive cells in GBM and PT areas. *N* = 50; the horizontal line represents the median value; *p* indicates the result of the Mann-Whitney test.

**Figure 3 ijms-22-01620-f003:**
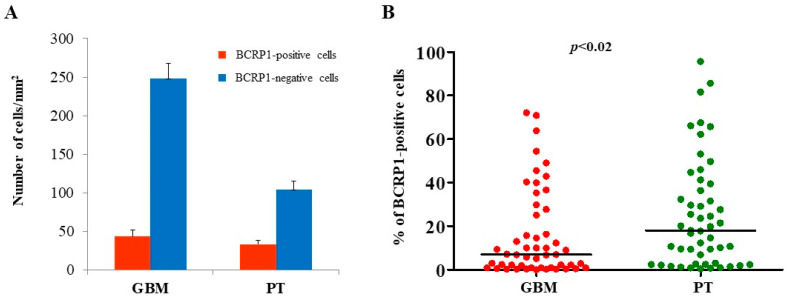
Stereological analysis of BCRP1 expression in both GBM and PT areas. Immunohistochemistry results were further evaluated by subjecting histological slides to an unbiased stereological analysis. Both BCRP1-labeled and unlabeled cells were counted, and the cell density of BCRP1 was graphed. (**A**) Red and blue bars show the number of BCRP1 cells/mm^2^. The data are presented as the mean ± SE. (**B**) Plots show the percentage of positive cells in GBM and PT areas. *N* = 50; the horizontal line represents the median value; *p* indicates the result of the Mann-Whitney test.

**Figure 4 ijms-22-01620-f004:**
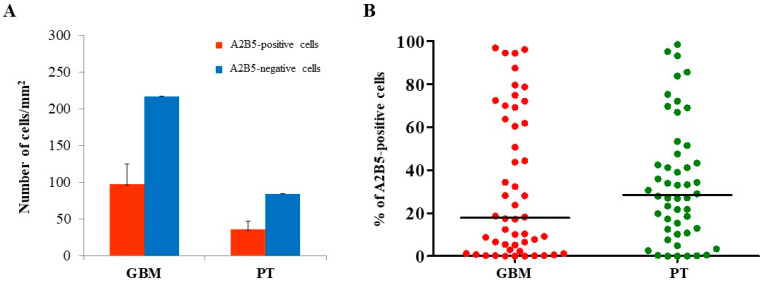
Stereological analysis of A2B5 expression in both GBM and PT areas. To more accurately evaluate the immunohistochemistry results, an unbiased stereological technique was used. Within each selected region of interest (ROI), the number of both immunopositive and immunonegative cells were used to estimate the expression density of A2B5 in the indicated histological areas. (**A**) Red and blue bars show the number of A2B5-positive and A2B5-negative cells/mm^2^, respectively. The data are presented as the mean ± SE. (**B**) Plots show the percentage of A2B5-positive cells in GBM and PT areas. *N* = 50; the horizontal line represents the median value; *p* indicates the result of the Mann-Whitney test.

**Figure 5 ijms-22-01620-f005:**
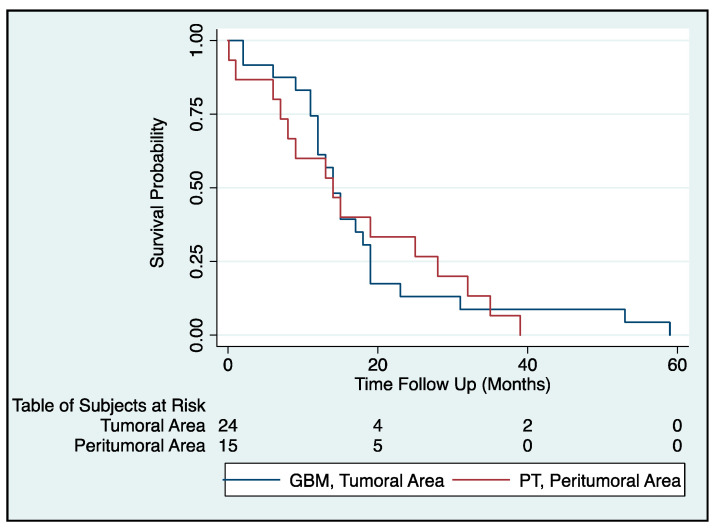
Survival of GBM patients is inversely correlated to MGMT expression in the PT area: The Kaplan–Meier plot depicts the differences in survival when patients with a percentage of MGMT-positive cells >50% were stratified based on the tissue areas (GBM, tumoral area; PT, peritumoral area). Furthermore, the number of surviving subjects per time unit is described in Table of Subjects at Risk. According to the Cox model (proportional hazard, Cox regression: Breslow method for ties), survival decreases significantly for patients showing >50% of MGMT-positive cells in the PT area (*p* < 0.05 log-rank test; *z* = −2.12; 95% CI = 0.026–0.00095).

**Table 1 ijms-22-01620-t001:** Clinicopathological characteristics of 55 patients with primary GBM

Variable		Total
Age at diagnosis	≤50	14
	>50	41
Gender	Male	34
	Female	21
Side	Left	17
	Right	28
	NA	10
Location	Frontal lobe	16
	Temporal lobe	9
	Parietal lobe	6
	Occipital lobe	4
	Others	25
Preoperative KPS score	≥80	52
	<80	3
Radiotherapy	yes	50
	no	5
Chemotherapy	yes	50
	no	5
Survival time (months)	Median	13
	Range	0–59
Clinical outcome	DOD	50
	DOOC	5

Abbreviations: KPS, Karnofsky performance status; DOD, dead of disease; DOOC, dead of other causes; NA, not assessed.

## Data Availability

The data presented in this study are available on request from the corresponding author.

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
