# Peer review of "Immunohistochemical Analysis of DNA Repair- and Drug-Efflux-Associated Molecules in Tumor and Peritumor Areas of Glioblastoma"

_ijms, 2021, doi:10.3390/ijms22041620_

Round 1

Reviewer 1 Report

In the manuscript of Angelucci et al, the authors investigated the expression of MGMT, BCRP1 and A2B5 in glioblastoma (GBM) and peritumoral tissues by immunohistochemistry. Since that these three molecules are involved in resistance and responsiveness to therapy, they proposed to identify a specific profile of the peritumoral zone able to explain the onset of resistance to treatment and recurrence. In addition, they suggested a correlation of the resulting peritumoral profile with clinical parameters.

Despite the aim of the study is interesting and the results are promising for the comprehension of the biology of GBM, the manuscript is too preliminary due to the lack of an in-depth investigation, which could effort the discovery of novel effective therapeutic strategies and the impact in the clinical setting.

Major concerns:

  • As mentioned at the end of the Discussion section, the authors should elucidate the significance of these data by performing additional experiments, such as coexpression analysis of markers of stemness, differentiation, angiogenesis-related factors, and then they could discuss and support their conclusions in the light of a more thorough investigation.
  • The authors should clarify how many patients were involved in the study. In the Results section they indicated 50 GBM patients, however in table 1 they reported information on 55 patients, as well as in Introduction, Materials and Methods and Discussion sections.
  • In the 2.2 paragraph of the Results section, the authors should describe more clearly the stratification groups. In particular, they should clarify the meaning of the variable histochemical group MGMTP cited in the text, whereas in the figure legend they defined the patient stratification based on the histopathological site as “GBM, site 0” and “PT area, site 1”. In addition, the results of Cox regression analysis to determine the prognostic value of BCRP1 and A2B5 in GBM patients should be shown as well, even if the results are not significant.

Minor concerns:

  • In the 2.1 paragraph of the Results section, the authors should describe more precisely the results observed for MGMT expression analysis, as they did for BCRP1 and A2B5. In addition, which statistical analysis did they perform for Figure 3? The descriptions in the text and in figure legend are different.
  • Figure legends are poorly written.

Author Response

Major concerns:

  • “As mentioned at the end of the Discussion section, the authors should elucidate the significance of these data by performing additional experiments, such as coexpression analysis of markers of stemness, differentiation, angiogenesis-related factors, and then they could discuss and support their conclusions in the light of a more thorough investigation.”

We agree with the Reviewer that additional co-expression analysis of the three investigated molecules with markers correlated to stemness, angiogenesis and differentiation would be useful in order to define expression profiles that may help to determine patients who may benefit or not from specific treatments. However, at present, we cannot perform these experiments since the amount of peritumoral tissue removed during surgery was very small and, for most of the patients included in this study, it is no longer available for further analysis. The current trend in gliomas’ surgery is to achieve a greater extent of tumour resection and novel techniques for accurate identification of peritumoural functional pathways and tumor cell infiltration are enabling surgeons to maximize cytoreduction. Nevertheless, the actual clinical benefit for patients depends on the balance between the degree of resection and the possibly onset of neurological morbidity (Sanai and Berger, 2018). Indeed, the aim of improving the oncological outcome must face the risks of neurological injuries (i.e., language dysfunction, motor deficits and neurocognitive impairment) related to removal of surrounding healthy tissue.

Moreover, as reported in the present paper, in our previous studies we have already detected the expression of stemness markers as well as differentiation and angiogenesis-related molecules in the same cohort of patient samples analysed in the present study. In all previous studies, molecule expression levels were compared between the two areas (GBM and peritumoral) (Mangiola et al., Clin Cancer Res 2007; 13: 6970;Sica et al. Int J Oncol 2011; 38: 41; Lama et al., J Neuropathol Exp Neurol 2016, 75: 134; D’Alessio et al. Oncotarget 2016; 7: 78541; Angelucci et al., Oncotarget 2018;9: 28116).

  • “The authors should clarify how many patients were involved in the study. In the Results section they indicated 50 GBM patients, however in table 1 they reported information on 55 patients, as well as in Introduction, Materials and Methods and Discussion sections.”

The patients enrolled in this study was 55. A lower number (50) has been reported in the result section since the limited amount of some tissue samples did not always allow us to perform the analysis of all the three molecules. Therefore, the expression of each of the three molecules was determined in 50 of 55 patients. We agree with the Reviewer that this information was probably not clearly deducible by reading the text, therefore we have added a sentence in paragraph 4.2 to clarify this issue.

  • “In the 2.2 paragraph of the Results section, the authors should describe more clearly the stratification groups. In particular, they should clarify the meaning of the variable histochemical group MGMTP cited in the text, whereas in the figure legend they defined the patient stratification based on the histopathological site as “GBM, site 0” and “PT area, site 1”. In addition, the results of Cox regression analysis to determine the prognostic value of BCRP1 and A2B5 in GBM patients should be shown as well, even if the results are not significant.”

We apologize with the Referee if this part of the manuscript was not clearly illustrated. We have replaced the Figure 5 with a new one. The legend of Figure 5 was also updated. The correct statistical analysis has been added to the text (see. Paragraph 2.1).

We have also included below two additional plots referring to the survival analysis related to BRCP1 and A2B5 molecules, respectively. These figures are not included in the revised manuscript.

BCRP1

Survival of patients is unconclusive correlated to BCRP1 expression in PT area: The Kaplan-Meier plot depicts the differences in survival when patients with a percentage of BCRP1 positive cells >50% were stratified based on the tissue areas (GBM, Tumoral Area; PT, Peritumoral Area). Furthermore, the number of surviving subjects per time unit is described in the "Table of Subjects at Risk". According to the Cox model (proportional hazard, Cox regression: Breslow method for ties), survival decreases for patients showing >50% of MGMT-positive cells in Tumoral area, but is not statistically significant. Log-rank test, p=0.651: z= -0.45; CI 95% (0.984-1.009).

A2B5

Survival of GBM patients is inversely correlated to A2B5 expression in PT area: The Kaplan-Meier plot depicts the differences in survival when patients with a percentage of A2B5 positive cells >50% were stratified based on the tissue areas (GBM, Tumoral Area; PT, Peritumoral Area). Furthermore, the number of surviving subjects per time unit is described in the "Table of Subjects at Risk". According to the Cox model (proportional hazard, Cox regression: Breslow method for ties), survival decreases but not statistically significant for patients showing >50% of A2B5-positive cells in PT area. Log-rank test, p=0.0896: z= 0.13; CI 95% (0.990-1.011).

Minor concerns:

  • “In the 2.1 paragraph of the Results section, the authors should describe more precisely the results observed for MGMT expression analysis, as they did for BCRP1 and A2B5. In addition, which statistical analysis did they perform for Figure 3? The descriptions in the text and in figure legend are different.”

The correct terminology referring to the statistical analysis has been indicated in the text as well as in the figures’ legends.

Reviewer 2 Report

MINOR

Discussion:

It is worthy to mention:

- the epigenetic interplay concerning MGMT expression. Particular miRNA molecules control MGMT activity. Cite adequate atricles.

- CSC connection with TLR-4 singaling wchich is crutial to determine  invasiveness.  

Cite:

Litak, J.; Grochowski, C.; Litak, J.; Osuchowska, I.; Gosik, K.; Radzikowska, E.; Kamieniak, P.; Rolinski, J. TLR-4 Signaling vs. Immune Checkpoints, miRNAs Molecules, Cancer Stem Cells, and Wingless-Signaling Interplay in Glioblastoma Multiforme—Future Perspectives. Int. J. Mol. Sci. 202021, 3114.

Author Response

Minor concerns:

  • “Discussion: It is worthy to mention:

- the epigenetic interplay concerning MGMT expression. Particular miRNA molecules control MGMT activity. Cite adequate articles.”

As suggested by the Reviewer, the role of miRNAs in control of MGMT expression/function has been included in the “Discussion” section and the related articles (new ref. 35) have been cited.

  • “CSC connection with TLR-4 singaling wchich is crutial to determine invasiveness.

Cite:

Litak, J.; Grochowski, C.; Litak, J.; Osuchowska, I.; Gosik, K.; Radzikowska, E.; Kamieniak, P.; Rolinski, J. TLR-4 Signaling vs. Immune Checkpoints, miRNAs Molecules, Cancer Stem Cells, and Wingless-Signaling Interplay in Glioblastoma Multiforme—Future Perspectives. Int. J. Mol. Sci. 2020, 21, 3114.”This is an extremely interesting and fascinating point. We are aware of the increased expression of some miRNAs in GBM tissue as well as in GMB cell lines. In addition, we believe that elucidating the mechanisms regulating the interplay between miRNA expression and TLR-related pathways may be crucial to understand the significance of immune inhibition in cancers, and in particular in GBM. However, since we did not address this topic in our manuscript, it would not be appropriate to cite the suggested article. Alternatively, we would have been forced to change the introduction and the discussions sections of our manuscript.

Round 2

Reviewer 1 Report

In the revised manuscript, the authors addressed all the concerns. Only one further correction is needed, as indicated in point 3) of Major concerns

Major Concerns:

  1. No further concerns
  2. The sentence added in the revised version clarified this point.
  3. The description of the stratification groups and the figure legend are clearly illustrated in the revised manuscript. However, the authors should also modify the terms used in the text accordingly to the new stratification groups (lane 178, revised version).

Minor concerns:

  • No further concerns